analytical chemistry/green chemistry/spectroscopy

synchronous spectrofluorimetry, febuxostat, ibuprofen, human plasma, green chemistry

**Author for correspondence:**
Galal Magdy
e-mail: galal_magdy@pharm.kfs.edu.eg

This article has been edited by the Royal Society of Chemistry, including the commissioning, peer review process and editorial aspects up to the point of acceptance.

# Two different synchronous spectrofluorimetric approaches for simultaneous determination of febuxostat and ibuprofen

Galal Magdy[1], Fathalla Belal[2], Ahmed M. Abdel-Megied[1,3] and Ahmed F. Abdel Hakiem[1]

[1]Pharmaceutical Analytical Chemistry Department, Faculty of Pharmacy, Kafrelsheikh University, P.O. Box 33511, Kafrelsheikh, Egypt
[2]Pharmaceutical Analytical Chemistry Department, Faculty of Pharmacy, Mansoura University, P.O. Box 35516, Mansoura, Egypt
[3]Department of Pharmaceutical Sciences, University of Maryland School of Pharmacy, 20N. Pine Street, Baltimore, MD 21201, USA

GM, 0000-0003-0543-5065

Two green, simple and sensitive synchronous spectrofluorimetric methods were developed for the first time for the simultaneous estimation of febuxostat (FEB) and ibuprofen (IBU). Method I is constant-wavelength synchronous spectrofluorimetry where FEB and IBU were recorded at 329 and 258 nm, respectively, using $\Delta\lambda$ of 40 nm. Method II is constant-energy synchronous spectrofluorimetry using a wavenumber interval of $-4000\ cm^{-1}$. All measurements were carried out in a borate buffer of pH 7 and distilled water for dilution which increased the methods' greenness. The two methods were rectilinear over concentration ranges of 30.0–700.0 ng ml$^{-1}$ and 0.5–9.0 µg ml$^{-1}$ in the first method and 20.0–500.0 ng ml$^{-1}$ and 0.1–8.0 µg ml$^{-1}$ in the second method for FEB and IBU, respectively. High sensitivity was attained for the two drugs with limits of quantitations (LODs) down to 0.41 and 5.51 ng ml$^{-1}$ in the first method and 0.25 and 3.32 ng ml$^{-1}$ in the second method for FEB and IBU, respectively. Recovery percentages were in the range of 97.3–101.9% after extraction from spiked human plasma samples, demonstrating high bioanalytical applicability. The two methods were further applied to tablet dosage forms with good recovery results. The methods' greenness was assessed according to the analytical Eco-Scale and Green Analytical Procedure Index guidelines.

# 1. Introduction

Gout is a type of inflammatory arthritis that is triggered by the deposition of monosodium urate crystals in the bones, joints and parenchymal organs such as the kidney. It is frequently associated with hyperuricemia [1]. This in turn results in nephrolithiasis, renal impairment and painful inflammatory arthritis. Therefore, gout management includes the treatment of gouty inflammation and hyperuricemia [2]. Several reports suggest that gout prevalence has increased in recent decades. The incidence of gout ranges from 0.1% to 10.0% through several regions worldwide and is increasing in a lot of developed countries [3]. The prevalence of gout in the US is 5.9% and 2.0% among men and women, respectively [4]. The UK had exhibited a rise to about 2.5% in 2012 [5].

Febuxostat (FEB) is a non-purine xanthine oxidase inhibitor that was approved by the FDA in 2009 for the treatment of chronic gout. Chemically, it is named as [2-[3-cyano-4-(2-methylpropoxy]phenyl]-4-methylthiazole-5-carboxylic acid] (figure 1a) [6]. Compared to allopurinol, FEB shows higher efficacy and tolerability without any significant allergic or drug–drug reactions, so that it is commonly prescribed for allopurinol intolerant patients [6,7].

Generally, treatment of gouty pain and the associated inflammation can be accomplished with corticosteroids, non-steroidal anti-inflammatory drugs (NSAIDs) and colchicine [2]. Colchicine and corticosteroids were reported to produce several side effects so that NSAIDs are the drugs of choice for the management of gouty flare and inflammation [8]. Accordingly, FEB is frequently co-administered with NSAIDs such as ibuprofen (IBU) [9].

IBU (2-[4-(2-methylpropyl) phenyl] propanoic acid) (figure 1b) is a NSAID that is usually prescribed for the management of rheumatic inflammation, musculoskeletal complaints and post-operative pain [6].

From the literature survey, the analysis of FEB was performed using different methods including spectrofluorimetry [10–13], spectrophotometry [14,15], HPLC [9,16–19], LC-MS [20–22], micellar electrokinetic capillary chromatography [23], HPTLC [24–27] and GC [28]. Similarly, the assay of IBU has been achieved applying different methods including spectrofluorimetry [29–32], spectrophotometry [33,34], HPLC [35–37], LC-MS [38,39], HPTLC [37], capillary zone electrophoresis [40,41], potentiometry [42] and GC [43,44].

Emission spectrofluorimetry is a low-cost technique characterized by high selectivity and sensitivity which permitted its application in the quantitative analysis of many pharmaceutical compounds. However, selectivity problems can arise during the simultaneous determination of multi-component samples with broad bands and overlapped emission spectra [45]. Therefore, the use of conventional emission spectrofluorimetry may include time-wasting pre-separation steps before analysis. By contrast, other techniques that include careful optimization of the instrumental parameters such as synchronous spectrofluorimetry (SS) and/or derivative SS allow the simultaneous determination of these mixtures by resolving such spectral overlap without any separation steps [31,46].

The synchronous spectrofluorimetric technique involves different modes; their principles, applications and procedures were reported [47]. Constant-wavelength synchronous spectrofluorimetry (CWSS) is the first mode, in which both excitation and emission monochromators are scanned concurrently while maintaining a constant difference ($\Delta\lambda$) between excitation ($\lambda_{ex}$) and emission ($\lambda_{em}$) wavelengths. The second mode is constant-energy synchronous spectrofluorimetry (CESS) in which the scanning is performed maintaining a constant-energy difference (i.e. wavenumber) between the monochromators [47].

To date, no synchronous spectrofluorimetric methods have been reported for concurrent analysis of FEB and IBU and this is considered the first analytical procedure for estimation of such a binary mixture.

Accordingly, the main objective of the present work is to develop and validate two green spectrofluorimetric synchronous methods for the analysis of FEB and IBU simultaneously by applying CWSS and CESS in methods I and II, respectively. The high sensitivity of the developed methods allowed their efficient application for estimation of the studied drugs in spiked human plasma samples and their tablets with high recovery percentages. Furthermore, the evaluation of the methods' greenness was performed according to analytical Eco-Scale and Green Analytical Procedure Index (GAPI) guidelines [48,49].

# 2. Experimental section

## 2.1. Reagents and materials

IBU (99.92%) was obtained from EIPICO Pharmaceutical Industries Company (10th of Ramadan City, Egypt). FEB was kindly provided by Eva-Pharm Company (Obour City, Egypt). Brufen® Tablets

**Figure 1.** Structural formulae of FEB (*a*) and IBU (*b*).

(600.0 mg IBU/tablet, Batch No. 08448-3 J-2-23, Kahira Pharmaceutical Company, Egypt) and Feburic® Tablets (80.0 mg FEB/tablet, Batch No. 1083001, Hikma Pharmaceuticals Co., Egypt) were purchased from a local pharmacy. A human plasma sample was obtained from Mansoura University Hospitals (Mansoura, Egypt).

Acetonitrile, methanol, ethanol, acetone, β-cyclodextrin, tween-80, carboxymethyl cellulose, sodium dodecyl sulfate, cetrimide, sodium hydroxide, boric acid, sodium acetate trihydrate and acetic acid 96.0% were provided by Sigma-Aldrich (St Louis, MO, USA).

Various pH values were obtained according to United States Pharmacopeia and acetic acid/sodium acetate buffer (pH 3.5–5.5) and 0.2 M of borate buffer (pH 6.5–10.0) were freshly prepared. In addition, surfactant solutions were also prepared with a concentration of 1.0% w/v or v/v.

## 2.2. Instrumentation

A Cary Eclipse Fluorescence Spectrometer from Agilent Technologies (Santa Clara, CA 95051, United States) and Cary Eclipse software were used for the measurement of synchronous fluorescence (SF) spectra at smoothing factor = 15 and slit width = 5 nm. The high voltage mode (800 V) of the instrument was applied and a 1.0 cm quartz cell was used for measurements. SF spectra were recorded at $\Delta\lambda = 40$ nm or a wavenumber interval = $-4000$ cm$^{-1}$ in methods I and II, respectively. Vortex mixer, model IVM-300p (Gemmy Industrial Corp, Taiwan), centrifuge, model 2-16P (Germany) and membrane filters with a pore size of 0.45 μm (Phenomenex, USA) were also used.

## 2.3. Standard solutions

Standard solutions of each of IBU and FEB (100.0 μg ml$^{-1}$) were prepared in methanol. Working standard solutions were obtained by further dilution of the standard solutions with the same solvent (50.0 μg ml$^{-1}$ for IBU and 40.0 μg ml$^{-1}$ for FEB). The obtained solutions were stored at 4°C and were found to be stable for at least two weeks.

## 2.4. General procedures

### 2.4.1. Construction of calibration curves

#### 2.4.1.1. Method I. constant-wavelength synchronous spectrofluorimetry
Aliquot volumes of standard solutions were moved into a group of 10.0 ml measuring flasks within a concentration range of 30.0–700.0 ng ml$^{-1}$ for FEB and 0.5–9.0 μg ml$^{-1}$ for IBU, followed by the addition of 2 ml volumes of borate buffer (0.2 M, pH 7). Each flask was completed to the volume with distilled water and mixed. Afterwards, SF spectra were recorded at 329 and 258 nm for FEB and IBU, respectively keeping a constant $\Delta\lambda$ of 40 nm between excitation and emission monochromators. Blank samples were measured similarly. The regression equations were derived by plotting the corrected SF intensities against the concentration of the drugs (ng ml$^{-1}$ or μg ml$^{-1}$) to obtain the calibration curves.

#### 2.4.1.2. Method II. constant-energy synchronous spectrofluorimetry
In this technique, the same procedure for CWSS was followed and SF spectra were measured in the range of 200–400 nm but keeping a constant wavenumber interval of $-4000$ cm$^{-1}$ between the monochromators. The SF spectra were recorded at 327 nm and 260 nm in the concentration range of 20.0–500.0 ng ml$^{-1}$ and 0.2–8.0 μg ml$^{-1}$ for FEB and IBU, respectively.

### 2.4.2. Analysis of febuxostat/ibuprofen synthetic mixtures

In methods I and II, different aliquots of each of the FEB and IBU working standard solutions were moved into 10.0 ml volumetric flasks to obtain five synthetic mixtures within each drug's linearity range. Volumes of 2 ml of borate buffer (0.2 M, pH 7) were added and the flasks were completed to volume with distilled water. Then we continued as mentioned under §'2.4.1.1' or '2.4.1.2' for methods I or II, respectively.

### 2.4.3. Analysis of febuxostat and ibuprofen in tablets

Ten Feburic® or Brufen® tablets were separately weighed and finely pulverized. An accurate amount equivalent to 600.0 mg of IBU or 80.0 mg of FEB was moved separately into 100.0 ml volumetric flasks, then about 40 ml of methanol was added. Sonication for 10 min was performed, and the flasks were completed with methanol to the volume followed by filtration. Increasing volumes of the filtrate were transferred into volumetric flasks (10.0 ml) then the steps mentioned under §'2.4.1.1' or '2.4.1.2' for methods I or II, respectively were applied. From the corresponding regression equations, the nominal contents of tablets were calculated.

### 2.4.4. Procedure for spiked human plasma

Into 15.0 ml centrifugation tubes set, aliquots of 1.0 ml of human plasma samples were transferred separately. Known volumes from stock solutions of FEB and IBU were added to the plasma to prepare five mixtures with different ratios (1 : 1, 1 : 2, 2 : 1, 1 : 3, and 1 : 10 for FEB:IBU) within the concentration range of each drug. Vortex for 3 min was carried out, and then precipitation of protein was performed by completing with methanol to 5 ml. Afterwards, the tubes were centrifuged at 5000 r.p.m. for 20 min. Aliquots of the supernatant (1.0 ml) were measured and filtered by syringe filters (0.45 µm) then moved into measuring flasks (10.0 ml). The drug concentrations were measured as described under §'2.4.1.1' or '2.4.1.2' for methods I or II, respectively in parallel with the blank experiment. The percentage recoveries of each drug and the regression equations were obtained by plotting the corrected SF intensities against the drug concentrations in ng ml$^{-1}$ or µg ml$^{-1}$.

## 3. Results and discussion

### 3.1. Spectral characteristics

As presented in figure 2, each of the FEB and IBU methanolic solutions show fluorescence at $\lambda_{em}$ of 389 and 287 nm following excitation at 313 and 260 nm, respectively. The SS technique was the best choice for the analysis of such a binary mixture in a single scan. The main advantages of SS included improved selectivity and sensitivity, light scattering reduction, and spectral simplification [50,51]. The SS results displayed a high level of tolerance to foreign substances particularly during the estimation of drugs in complicated biological matrices and dosage forms. Moreover, the analysis of the drugs could be easily achieved by SS due to its sharp and narrow spectrum [46]. Accordingly, two green and simple synchronous spectrofluorimetric methods were developed applying CWSS and CESS in methods I and II, respectively, for concurrent analysis of FEB and IBU for the first time without any pre-separation steps.

### 3.2. Constant-wavelength synchronous spectrofluorimetry

Different $\Delta\lambda$ settings were carefully investigated in the range of 10–100 nm. The maximum fluorescence intensity for FEB was obtained at $\Delta\lambda = 80$ nm which is equivalent to the difference between its $\lambda_{ex}$ and $\lambda_{em}$ (electronic supplementary material, figure S1a) [52], while for IBU, $\Delta\lambda = 30$ nm produced the highest fluorescence intensity (electronic supplementary material, figure S1b). To obtain the SF spectra in a single scan for the two drugs with a reasonable sensitivity, $\Delta\lambda = 40$ nm was found to be appropriate and produced a reasonable response for both drugs (electronic supplementary material, figure S1c). The intensities of SF spectra were recorded at 329 nm for FEB and 258 nm for IBU (figure 3). Well resolved peaks were obtained which permitted the determination of the two drugs in the presence of each other without interference.

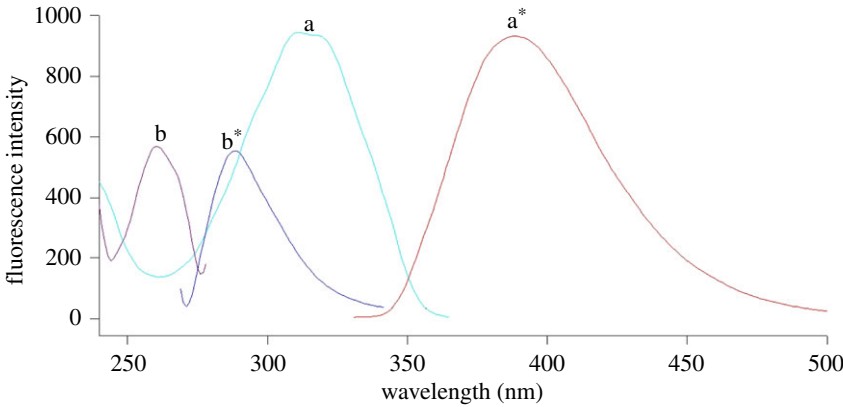

**Figure 2.** Excitation and emission spectra of FEB (400.0 ng ml$^{-1}$) (a, a*), and IBU (4.0 µg ml$^{-1}$) (b, b*).

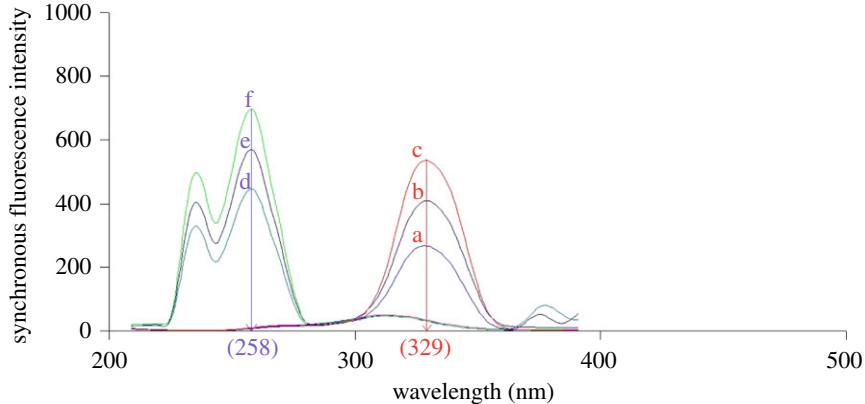

**Figure 3.** Synchronous fluorescence spectra of the studied drugs at $\Delta\lambda = 40$ nm. Where (a,b,c): 200.0, 300.0 and 400.0 ng ml$^{-1}$ FEB, (d,e,f): 4.0, 5.0 and 6.0 µg ml$^{-1}$ IBU.

In this technique, the SF spectra of various concentrations of FEB were recorded at 329 nm with a fixed concentration (5.0 µg ml$^{-1}$) of IBU (figure 4a), while the SF spectra of IBU were recorded at 258 nm with a fixed concentration (600.0 ng ml$^{-1}$) of FEB (figure 4b).

## 3.3. Constant-energy synchronous spectrofluorimetry

The constant-energy mode was the second mode used to obtain the SF spectra. Wavenumber intervals ranging from −500 to −10 000 cm$^{-1}$ were investigated, and it was found that −4000 cm$^{-1}$ was appropriate for determining both drugs with good resolution and sensitivity in a single scan (electronic supplementary material, figure S2). In this method, the intensities of SF spectra were recorded at 327 nm and 260 nm for FEB and IBU, respectively (figure 5).

The SF spectra of various concentrations of FEB were recorded at 327 nm with a fixed concentration (5.0 µg ml$^{-1}$) of IBU as illustrated in figure 6a. Meanwhile, those of IBU were recorded at 260 nm with a fixed concentration (300.0 ng ml$^{-1}$) of FEB (figure 6b).

## 3.4. Optimization of experimental parameters

### 3.4.1. Effect of diluting solvents

The influence of diluting solvents on SF intensities was tested using distilled water, acetone, acetonitrile, methanol and ethanol. In both methods, distilled water produced the highest SF intensities for both FEB and IBU with high resolution. On the other hand, acetonitrile, methanol, ethanol and acetone reduced significantly the SF intensities of the two drugs (electronic supplementary material, figure S3a). Thus,

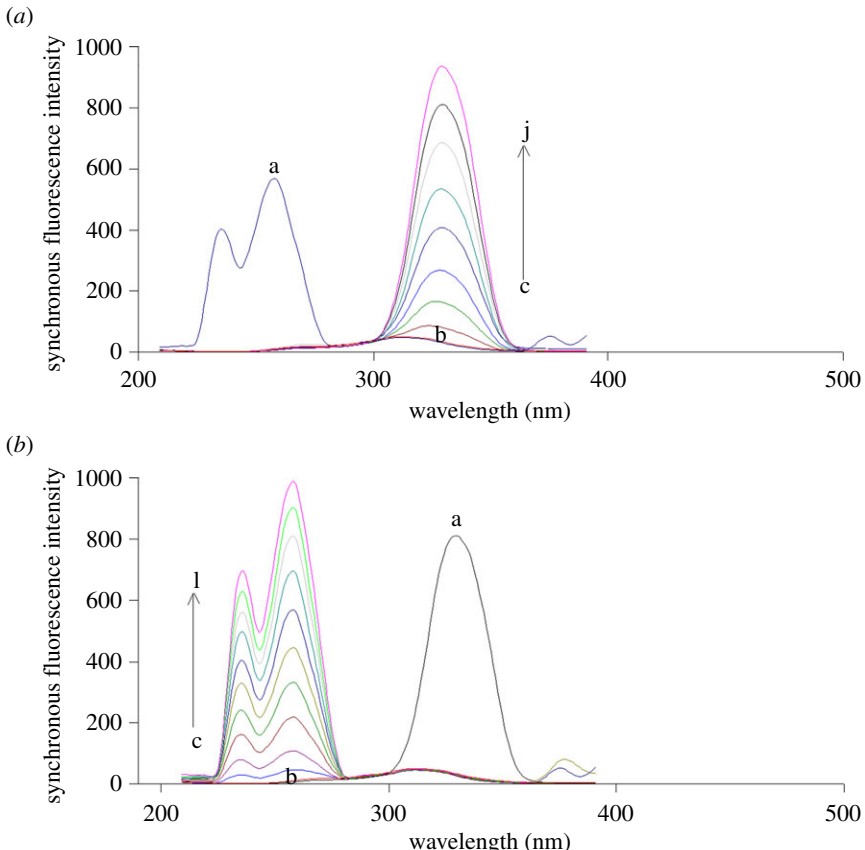

**Figure 4.** Synchronous fluorescence spectra of (*a*) (a) IBU (5.0 µg ml$^{-1}$), (b) blank and (c–j: 30.0, 100.0, 200.0, 300.0, 400.0, 500.0, 600.0, 700.0 ng ml$^{-1}$ FEB) at $\Delta\lambda = 40$ nm, (*b*) (a) FEB (600.0 ng ml$^{-1}$), (b) blank and (c–l: 0.5, 1.0, 2.0, 3.0, 4.0, 5.0, 6.0, 7.0, 8.0, 9.0 µg ml$^{-1}$ IBU) at $\Delta\lambda = 40$ nm.

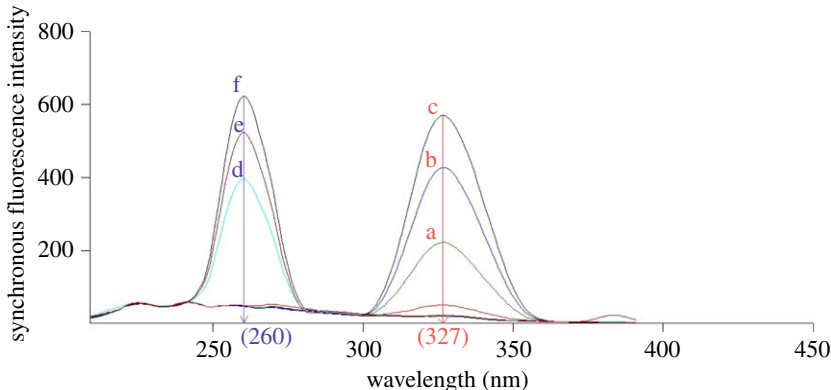

**Figure 5.** Synchronous fluorescence spectra of the studied drugs at wavenumber interval $= -4000$ cm$^{-1}$. Where (a,b,c): 200.0, 300.0 and 400.0 ng ml$^{-1}$ FEB, (d,e,f): 4.0, 5.0 and 6.0 µg ml$^{-1}$ IBU.

the best diluting solvent used in both methods was distilled water, which increased the greenness of the proposed methods.

### 3.4.2. Effect of pH

Borate and acetate buffers (0.2 M) with pH ranging from 6.5 to 10.0 and 3.5 to 5.5, respectively, were examined. After several trials, it was found that the highest SF intensity for each of FEB and IBU was obtained at pH 7 using 0.2 M borate buffer. The highly basic or acidic pH values didn't result in a significant improvement of SF intensities (electronic supplementary material, figure S3b).

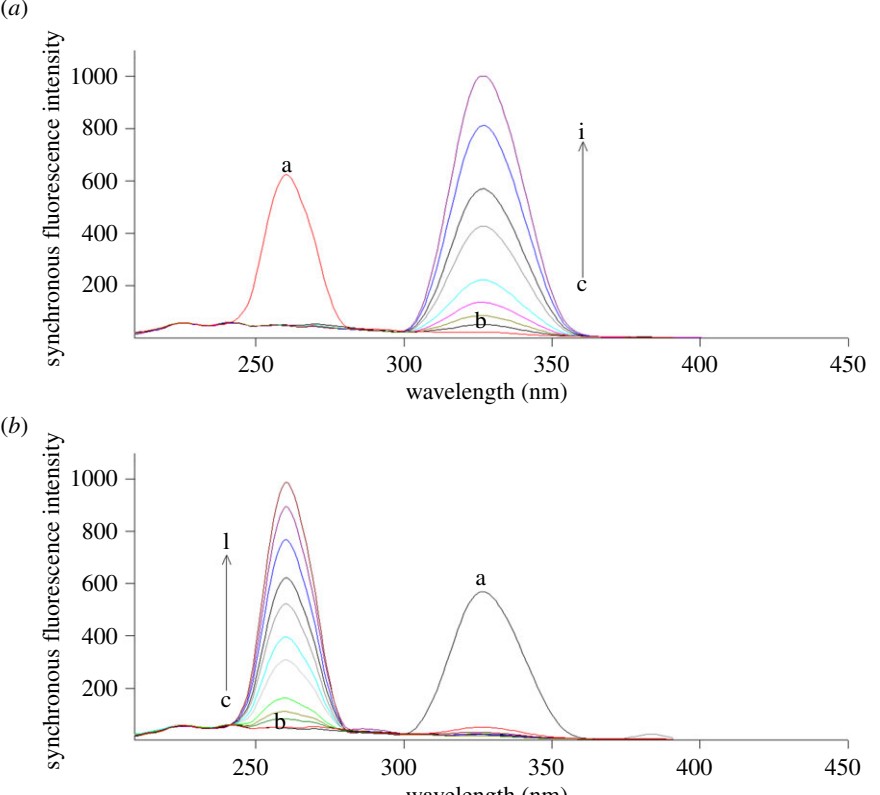

**Figure 6.** Synchronous fluorescence spectra of: (*a*) (a) IBU (5.0 µg ml$^{-1}$), (b) blank and (c–i: 20.0, 50.0, 100.0, 200.0, 300.0, 400.0, 500.0 ng ml$^{-1}$ FEB), (*b*) (a) FEB (300.0 ng ml$^{-1}$), (b) blank and (c–l: 0.2, 0.5, 1.0, 2.0, 3.0, 4.0, 5.0, 6.0, 7.0, 8.0 µg ml$^{-1}$ IBU) at wavenumber interval = −4000 cm$^{-1}$.

Consequently, different volumes of borate buffer (0.5–3.0 ml) were studied. In both methods, the optimum SF intensities were obtained using 2 ml of borate buffer (electronic supplementary material, figure S3c).

### 3.4.3. Effect of organized media

Different organized media were tested for their probable ability to improve the SF intensities of cited drugs in concentrations more than their critical micelle concentrations [53]. The studied organized media include surfactants such as cetrimide, sodium dodecyl sulfate and tween-80, as well as macromolecules such as carboxy methyl cellulose, and β-cyclodextrin. In the two methods, no significant improvement of the SF intensity of the cited drugs was obtained by any of the mentioned organized media (electronic supplementary material, figure S3d). Thus, no organized medium was used in this study.

## 3.5. Method validation

Validation of the proposed methods was performed in accordance with ICHQ2(R1) regulations [54].

### 3.5.1. Linearity and range

A linear relationship was obtained by plotting drug concentrations in ng ml$^{-1}$ or µg ml$^{-1}$ against corrected SF intensities calculated from the two fluorescence spectra (CWSS and CESS) for both drugs. The obtained concentration ranges are abridged in table 1.

The high values of the correlation coefficients ($r > 0.999$) with relatively small values of intercepts and the statistical analysis [55] showed that the calibration graphs are satisfactorily linear (table 1).

### 3.5.2. Limits of quantitation and detection

The LOD and LOQ values for each drug in both methods were determined and are summarized in table 1. Concerning the methods' sensitivity, the values of LOD and LOQ showed that the suggested methods can analyse the cited drugs with high sensitivity down to nanogram levels, and so can be efficiently applied for their analysis at the biological level.

### 3.5.3. Accuracy and precision

The results of the proposed methods were statistically compared with those acquired by the comparison methods [11,31], and insignificant differences were found between the performances of the two methods concerning accuracy and precision as shown in table 2. This was proved by statistical analysis of the data using student *t*-test and variance ratio *F*-test, respectively [55]. In addition, a high percentage of recoveries (98.91–101.14%) was obtained indicating good accuracy of the developed methods. The small values of % RSD (less than 2%) also indicated adequate precision of the two methods. The intra-day and inter-day precisions were examined and low % error and % RSD values were obtained, demonstrating the good precision of the developed methods (electronic supplementary material, table S1).

### 3.5.4. Robustness

The minor changes that could affect the SF intensities of the two drugs were evaluated to study the methods' robustness. The effects of volume of borate buffer (2 ml ± 0.1) and pH (7 ± 0.1) were investigated and insignificant effects on the values of % RSD and % recoveries were obtained in both methods (electronic supplementary material, table S2).

### 3.5.5. Selectivity

Both methods were used for the concurrent analysis of FEB and IBU with high recoveries and without interference. Moreover, the selectivity of both methods was also proved through the estimation of the cited drugs in complicated matrices of spiked human plasma samples. It was found that the two methods have enough selectivity to analyse the cited drugs with high % recoveries (97.3–101.9%) and small % RSD values, indicating no interference from plasma endogenous components.

## 3.6. Applications

### 3.6.1. Analysis of febuxostat/ibuprofen synthetic mixtures

The suggested methods were applied for the estimation of synthetic mixtures of different ratios of FEB and IBU as presented in figure 7. Using the corresponding regression equations, the concentrations of the cited drugs in their synthetic mixtures could be determined. The obtained results demonstrated the method accuracy as presented in table 3.

### 3.6.2. Analysis of the studied drugs in their dosage forms

The proposed methods were efficiently used for the estimation of FEB and IBU in their tablets (Feburic® and Brufen®) without interference from co-formulated excipients. The proposed and comparison methods [11,31] were statistically compared using the variance ratio *F*-test and student's *t*-test [55], and there were no significant variations between them in terms of precision and accuracy, respectively, as shown in table 4.

### 3.6.3. Analysis of febuxostat and ibuprofen in spiked human plasma samples

Simultaneous analysis of FEB and IBU was carried out in spiked human plasma samples with respect to their biological concentrations [56,57]. The maximum plasma concentration ($C_{max}$) of IBU was reported to be 21–150 µg ml$^{-1}$ (mean, 83) in 1–3 h when it is administered as a single dose of 20–30 mg kg$^{-1}$ [56], while the $C_{max}$ of FEB was reported as 0.65 µg ml$^{-1}$ upon administration of 40 mg daily [57]. The high sensitivity of the developed methods down to 1.26 and 16.71 ng ml$^{-1}$ in method I, and 0.76 and 10.12 ng ml$^{-1}$ in method II for FEB and IBU, respectively, permitted the assay of the two drugs at a biological level. When the SF intensities were plotted against the drug concentrations (µg ml$^{-1}$ or

**Table 1.** Analytical performance data for the proposed methods.

| parameters | method I (CWSS) | | | | method II (CESS) | | |
|---|---|---|---|---|---|---|---|
| | FEB | | IBU | | FEB | | IBU |
| wavelength (nm) | 329 nm | | 258 nm | | 327 nm | | 260 nm |
| | ($\Delta\lambda = 40$ nm) | | ($\Delta\lambda = 40$ nm) | | (wavenumber interval = $-4000$ cm$^{-1}$) | | (wavenumber interval = $-4000$ cm$^{-1}$) |
| linearity range | 30.0–700.0 | | 0.5–9.0 | | 20.0–500.0 | | 0.2–8.0 |
| | (ng ml$^{-1}$) | | (µg ml$^{-1}$) | | (ng ml$^{-1}$) | | (µg ml$^{-1}$) |
| intercept ($a$) | 14.17 | | −8.46 | | 24.91 | | 46.14 |
| slope ($b$) | 1.32 | | 113.71 | | 1.95 | | 118.61 |
| correlation coefficient ($r$) | 0.9999 | | 0.9999 | | 0.9999 | | 0.9999 |
| s.d. of the residuals, $S_{y/x}$ | 0.91 | | 0.96 | | 1.29 | | 0.86 |
| s.d. of the intercept, $S_a$ | 0.16 | | 0.19 | | 0.15 | | 0.12 |
| s.d. of the slope, $S_b$ | 0.001 | | 0.109 | | 0.003 | | 0.103 |
| percentage relative standard deviation, % RSD | 0.61 | | 0.33 | | 0.55 | | 0.68 |
| percentage relative error, % error | 0.20 | | 0.11 | | 0.21 | | 0.21 |
| limit of detection, LOD[a] (ng ml$^{-1}$) | 0.41 | | 5.51 | | 0.25 | | 3.32 |
| limit of quantitation, LOQ[b] (ng ml$^{-1}$) | 1.26 | | 16.71 | | 0.76 | | 10.12 |

[a]LOD = 3.3 $S_a/b$.
[b]LOQ = 10 $S_a/b$, where $S_a$ = standard deviation of the intercept and $b$ = slope.

**Table 2.** Application of the proposed methods for the determination of the studied drugs in raw materials.

| parameter | method I (CWSS) | | | | method II (CESS) | | | |
| | FEB | | IBU | | FEB | | IBU | |
| | amount taken (ng ml$^{-1}$) | mean$^a$ ± s.d. | amount taken (µg ml$^{-1}$) | mean$^a$ ± s.d. | amount taken (ng ml$^{-1}$) | mean$^a$ ± s.d. | amount taken (µg ml$^{-1}$) | mean$^a$ ± s.d. |
|---|---|---|---|---|---|---|---|---|
| | 30.0 | 100.45 ± 0.71 | 0.5 | 99.19 ± 0.41 | 20.0 | 99.86 ± 0.48 | 0.2 | 100.59 ± 0.84 |
| | 50.0 | 101.14 ± 0.46 | 1.0 | 99.24 ± 0.37 | 50.0 | 99.21 ± 0.56 | 0.5 | 100.94 ± 0.69 |
| | 100.0 | 100.51 ± 0.56 | 2.0 | 99.71 ± 0.52 | 100.0 | 100.70 ± 0.49 | 1.0 | 100.22 ± 0.58 |
| | 200.0 | 100.20 ± 0.63 | 3.0 | 99.27 ± 0.44 | 200.0 | 99.66 ± 0.61 | 2.0 | 99.43 ± 0.81 |
| | 300.0 | 99.46 ± 0.59 | 4.0 | 99.94 ± 0.27 | 300.0 | 99.65 ± 0.52 | 3.0 | 99.45 ± 0.70 |
| | 400.0 | 98.91 ± 0.83 | 5.0 | 100.34 ± 0.31 | 400.0 | 100.80 ± 0.47 | 4.0 | 100.52 ± 0.55 |
| | 500.0 | 100.77 ± 0.75 | 6.0 | 100.46 ± 0.43 | 500.0 | 99.65 ± 0.65 | 5.0 | 99.13 ± 0.81 |
| | 600.0 | 100.37 ± 0.51 | 7.0 | 100.55 ± 0.34 | | | 6.0 | 100.88 ± 0.52 |
| | 700.0 | 99.76 ± 0.64 | 8.0 | 100.61 ± 0.29 | | | 7.0 | 100.44 ± 0.60 |
| | | | 9.0 | 98.99 ± 0.48 | | | 8.0 | 99.48 ± 0.58 |
| | comparison method [11] | | comparison method [31] | | | | | |
| mean ± s.d. | 99.43 ± 0.74 | | 99.79 ± 1.12 | | | | | |
| $N^c$ | 3 | | 3 | | | | | |
| t-test | 0.63 (2.36)$^b$ | | 0.87 (2.36)$^b$ | | | | | |
| F-value | 1.94 (19.29)$^b$ | | 2.42 (19.29)$^b$ | | | | | |

$^a$Mean of three determinations.

$^b$The values between parentheses are the tabulated t and F values at p = 0.05 [55].

$^c$Number of samples.

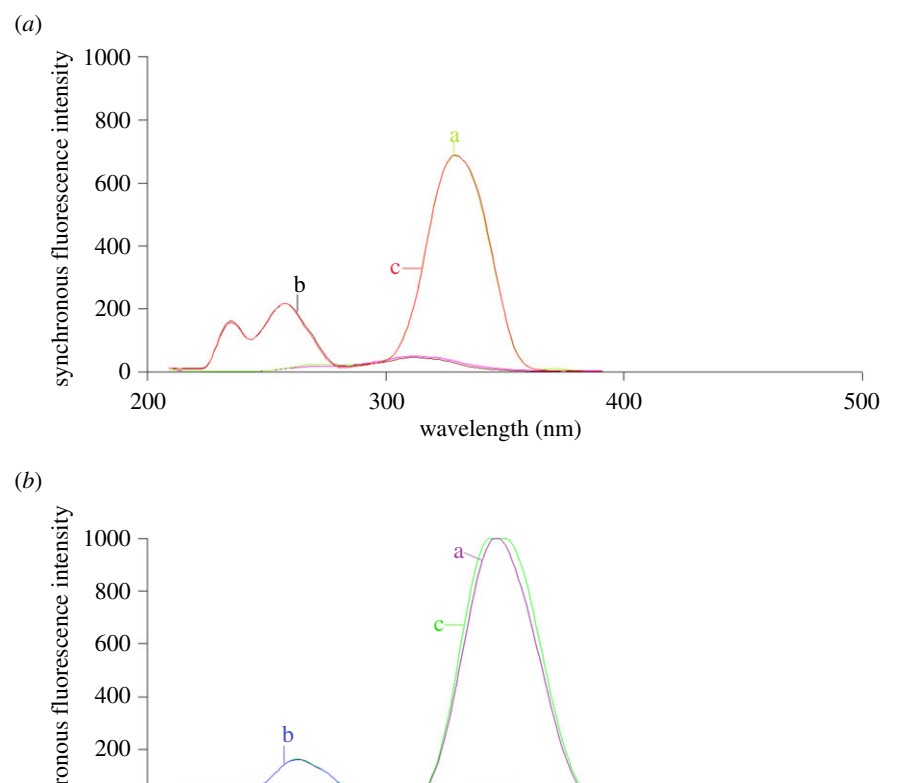

**Figure 7.** Synchronous fluorescence spectra of (*a*) (a) FEB (500.0 ng ml$^{-1}$), (b) IBU (2.0 µg ml$^{-1}$), (c) synthetic mixture of FEB/IBU at $\Delta\lambda = 40$ nm (method I), (*b*) (a) FEB (500.0 ng ml$^{-1}$), (b) IBU (1.0 µg ml$^{-1}$), (c) synthetic mixture of FEB/IBU at wavenumber interval $= -4000$ cm$^{-1}$ (method II).

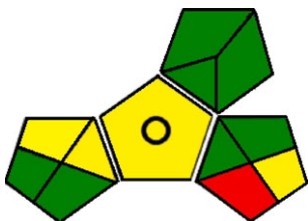

**Figure 8.** GAPI assessment of the green profile of the proposed methods.

ng ml$^{-1}$) in spiked samples with IBU and FEB, a linear relationship was obtained (table 5). The developed methods were found to have high % recoveries (97.3–101.9%) and low % RSD values, showing that they could be used to estimate the cited drugs in spiked human plasma with high selectivity.

## 3.7. Evaluation of greenness of the proposed methods

Both GAPI and analytical Eco-Scale tools have been used to assess the greenness of the proposed methods. GAPI guidelines [49] were followed and as shown in figure 8, the proposed methods are direct. The two methods also produce a low amount of waste and require low amounts of non-toxic substances. Additionally, the proposed methods are for qualification and quantification. Concerning analytical Eco-Scale, a numerical score is calculated which demonstrates the method's greenness [48]. From the total of 100 points, penalty points are deducted for each harmful effect on the environment. Taking into account the hazard pictograms during the calculation, the Eco-Scale total score was equal to 88 for both methods demonstrating that the developed methods are excellent green methods (table 6).

**Table 3.** Assay results for the determination of FEB and IBU in synthetic mixtures using the proposed methods.

| mix. number | method I (CWSS) | | | | | | method II (CESS) | | | | | |
|---|---|---|---|---|---|---|---|---|---|---|---|---|
| | amount taken | | mean$^a$ ± s.d. | | | | amount taken | | mean$^a$ ± s.d. | | | |
| | (ng ml$^{-1}$) | (µg ml$^{-1}$) | | FEB | | IBU | (ng ml$^{-1}$) | (µg ml$^{-1}$) | | FEB | | IBU |
| | FEB | IBU | | FEB | | IBU | FEB | IBU | | FEB | | IBU |
| 1 | 700.0 | 0.7 | | 99.37 ± 0.85 | | 100.04 ± 0.65 | 500.0 | 0.5 | | 99.40 ± 1.1 | | 98.73 ± 0.83 |
| 2 | 500.0 | 1.0 | | 100.58 ± 0.68 | | 99.85 ± 0.44 | 500.0 | 1.0 | | 99.86 ± 0.91 | | 100.57 ± 0.92 |
| 3 | 600.0 | 1.8 | | 100.84 ± 0.89 | | 99.48 ± 0.53 | 400.0 | 0.2 | | 101.19 ± 0.84 | | 99.42 ± 0.93 |
| 4 | 500.0 | 2.0 | | 100.04 ± 0.77 | | 100.72 ± 0.59 | 300.0 | 0.9 | | 99.05 ± 0.98 | | 100.54 ± 1.04 |
| 5 | 400.0 | 4.0 | | 99.06 ± 0.79 | | 99.99 ± 0.49 | 400.0 | 4.0 | | 100.23 ± 0.87 | | 100.01 ± 0.80 |
| 6 | | | | | | | 500.0 | 2.0 | | 100.17 ± 0.95 | | 99.93 ± 0.91 |

$^a$Mean of three determinations.

**Table 4.** Determination of FEB and IBU in tablet dosage forms using the proposed methods.

| preparation | method I (CWSS) | | method II (CESS) | |
|---|---|---|---|---|
| | amount taken (ng ml$^{-1}$) | mean$^a$ ± s.d. | amount taken (ng ml$^{-1}$) | mean$^a$ ± s.d. |
| Feburic® tablets FEB (80.0 mg per tablet) | 100.0 | 100.32 ± 0.55 | 100.0 | 100.68 ± 0.81 |
| | 200.0 | 100.24 ± 0.78 | 200.0 | 99.37 ± 0.57 |
| | 300.0 | 99.15 ± 0.59 | 300.0 | 100.46 ± 0.66 |
| | 400.0 | 100.60 ± 0.63 | 400.0 | 99.57 ± 0.61 |
| | 500.0 | 99.87 ± 0.68 | 500.0 | 100.18 ± 0.59 |
| | comparison method [11] | | | |
| mean ± s.d. | 99.39 ± 0.84 | | | |
| $N^c$ | 3 | | | |
| *t*-test | 0.41 (2.36)$^b$ | | | |
| *F*-value | 1.63 (19.29)$^b$ | | | |

| preparation | method I (CWSS) | | method II (CESS) | |
|---|---|---|---|---|
| | amount taken (µg mL$^{-1}$) | mean$^a$ ± s.d. | amount taken (µg mL$^{-1}$) | mean$^a$ ± s.d. |
| Brufen® tablets IBU (600.0 mg per tablet) | 1.0 | 99.32 ± 0.61 | 1.0 | 99.22 ± 0.69 |
| | 2.0 | 99.66 ± 0.73 | 2.0 | 100.47 ± 0.73 |
| | 3.0 | 100.34 ± 0.70 | 3.0 | 100.62 ± 0.88 |
| | 4.0 | 100.68 ± 0.52 | 4.0 | 99.15 ± 0.79 |
| | 5.0 | 99.52 ± 0.56 | 5.0 | 100.28 ± 0.85 |
| | comparison method [31] | | | |
| mean ± s.d. | 99.34 ± 1.24 | | | |
| $N^c$ | 3 | | | |
| *t*-test | 0.75 (2.36)$^b$ | | | |
| *F*-value | 2.41 (19.29)$^a$ | | | |

$^a$Mean of three determinations.

$^b$The values between parentheses are the tabulated *t* and *F* values at *p* = 0.05 [55].

$^c$Number of samples.

**Table 5.** Application of the proposed methods for the determination of the studied drugs in spiked human plasma.

| | method I (CWSS) | | | | method II (CESS) | | | |
| | amount added | | mean[a] ± s.d. | | amount added | | mean[a] ± s.d. | |
| | (ng ml⁻¹) | (µg ml⁻¹) | | | (ng ml⁻¹) | (µg ml⁻¹) | | |
| mix. number | FEB | IBU | FEB | IBU | FEB | IBU | FEB | IBU |
|---|---|---|---|---|---|---|---|---|
| 1 | 700.0 | 0.7 | 100.60 ± 1.71 | 101.11 ± 1.97 | 500.0 | 0.5 | 98.90 ± 1.35 | 101.90 ± 1.18 |
| 2 | 500.0 | 1.0 | 97.91 ± 1.87 | 97.27 ± 1.85 | 500.0 | 1.0 | 100.44 ± 1.42 | 99.27 ± 1.45 |
| 3 | 600.0 | 1.8 | 99.78 ± 1.21 | 99.77 ± 1.91 | 400.0 | 0.2 | 100.43 ± 0.86 | 98.85 ± 1.33 |
| 4 | 500.0 | 2.0 | 100.10 ± 1.42 | 101.63 ± 1.98 | 300.0 | 0.9 | 98.90 ± 1.12 | 99.91 ± 1.57 |
| 5 | 400.0 | 4.0 | 101.77 ± 1.66 | 99.78 ± 1.82 | 400.0 | 4.0 | 101.23 ± 1.07 | 100.03 ± 1.13 |

[a]Mean of three determinations.

**Table 6.** The penalty points of the proposed methods according to the analytical Eco-Scale per sample.

| reagents | penalty points |
| --- | --- |
| borate buffer (0.2 M) | 0 |
| methanol (900 µl) | 6 |
| *instrument* | *penalty points* |
| spectrofluorimeter (less than 0.1 kWh per sample) | 0 |
| occupational hazard (analytical process hermetization) | 0 |
| waste (10 ml, no treatment) | 6 |
| *total penalty points* | 12 |
| *analytical eco-scale total score*[a] | 88 |

[a]If the score is greater than 75, it represents excellent green analysis. If the score is greater than 50, it represents acceptable green analysis. If the score is less than 50, it represents inadequate green analysis.

## 3.8. Comparison between the two developed methods

It is clear from the above discussion that both CWSS and CESS techniques could be used for estimation of the cited drugs with high sensitivity down to nanogram levels. In comparison between the two developed methods, the CESS offered an improvement in sensitivity and selectivity compared to the more conventional CWSS technique, which was in agreement with the reported literature [58]. This can be attributed to the reduction of solvent Raman scatter and Rayleigh scatter interferences with the fluorescence spectra of the analytes [59]. This effect is higher in the case of the CESS technique when compared to CWSS, which could be useful especially in the case of weakly fluorescent analytes or low concentrations of strongly fluorescent compounds [60]. Consequently, the current study applied both techniques to obtain the highest sensitivity for both drugs.

# 4. Conclusion

Two eco-friendly and simple spectrofluorimetric methods were developed applying each of CWSS and CESS for the determination of a mixture of FEB and IBU without any separation steps. The proposed methods introduce the first spectroscopic approaches for the estimation of such a binary mixture. The proposed methods exhibited high sensitivity down to nanogram levels which allowed their application for the determination of investigated drugs with high per cent of recoveries in human plasma. Moreover, the proposed methods are excellent green, simple, rapid and economic methods without the need for sophisticated instruments. Additionally, the developed methods permit analysis of the cited drugs with short analysis time, minimum sample preparation and they also offer wide linearity ranges. The methods' greenness has been successfully assessed using GAPI and analytical Eco-Scale guidelines. Full validation of the developed methods was performed according to the ICH regulations.

Ethics. The collection of biological samples has been reviewed and approved by the ethical committee of the Faculty of Pharmacy, Kafrelsheikh University (code no. 2020-4).

Data accessibility. Data are available at the Dryad Digital Repository: https://doi.org/10.5061/dryad.r4xgxd2bv [61].

Authors' contributions. G.M. carried out the laboratory work, participated in data analysis and drafted the manuscript; A.M.A. and A.F.A. participated in the design of the study, the statistical analysis and revised the manuscript; F.B. organized the overall study and revised the manuscript. All authors approved the manuscript for publication.

Competing interests. The authors declare that they have no competing interests.

Funding. No funding supported this research.

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
