## [Peer Review File · Royal Society Open Science]

Review History

RSOS-210354.R0 (Original submission)

Review form: Reviewer 1

Is the manuscript scientifically sound in its present form?

Yes

Are the interpretations and conclusions justified by the results?

Yes

Is the language acceptable?

Yes

Do you have any ethical concerns with this paper?

No

Have you any concerns about statistical analyses in this paper?

No

Recommendation?

Accept with minor revision (please list in comments)

Comments to the Author(s)

In this study, the authors developed two green and sensitive synchronous spectrofluorometric methods (CWSS and CESS) for simultaneous determination of febuxostat and ibuprofen. They applied the proposed methods to pharmaceuticals and human plasma. It is shown how green the methods are using two different greenness assessment tools; GAPI and analytical Eco-Scale. I think the developed methods are useful and have advantages over the published ones. The manuscript is well written and the experimental strategies were explained in detail. The context of the subject in the literature is demonstrated and the paper is easy to understand. I see this manuscript is acceptable for publication after few minor changes. The suggestions were listed below.

1. The manuscript should be rechecked and any typos have to be corrected.
2. In abstract: line number 33, "Furthermore" may be removed.
3. In section 2.2, the used cell should be written.
4. In section 3.2: line number 18, change "The intensities of SF" into "The intensities of SF spectra".
5. Reference number 60, authors' names need corrections and the title should be consistent with that in Dryad Digital Repository.
6. Table 1: add superscript letters on LOD and LOQ in the table and its footnotes.
7. Table 4: change "pharmaceutical preparations" into "tablet dosage forms" in the caption.

Review form: Reviewer 2 (Mahmoud A. Omar)

Is the manuscript scientifically sound in its present form?

Yes

Are the interpretations and conclusions justified by the results?

Yes

Is the language acceptable?

Yes

Do you have any ethical concerns with this paper?

No

Have you any concerns about statistical analyses in this paper?

No

Recommendation?

Major revision is needed (please make suggestions in comments)

Comments to the Author(s)

The manuscript presents two synchronous spectrofluorimetric approaches for the simultaneous analysis of combined drugs; febuxostat and ibuprofen. The developed methods were applied to pharmaceutical tablets and human plasma. I recommend that the proposed methods are green and sensitive enough for simultaneous estimation of such a mixture. The manuscript is well-written and the topic is interesting. It can be accepted for publication in Royal Society Open Science after performing the required revision.

Comments:

1. Please revise the adjustment of the text within the manuscript to be uniform.

2. In the abstract; "the method greenness" in lines no. 17 and 36 should be corrected into "the methods' greenness".
3. In the Introduction section; the first sentence "Gout is a type of inflammatory arthritis that is triggered by the deposition of monosodium urate crystals in the bones, joints, and parenchymal organs such as the kidney and is frequently associated with hyperuricemia" should be shorter.
4. Section 2.1; the second sentence, change "FEB" into "Febuxostat" so that it doesn't start with an abbreviation.
5. Section 2.2; please add other instruments used for centrifugation, vortex, and filtration.
6. Section 2.4.3; please explain why 10 min sonication is necessary.
7. Section 3.7; line no. 16, "direct methods", remove "methods" as it is a redundant word.
8. Please revise the significant figures all over the manuscript and they should be unified.
9. Please revise the manuscript for grammatical and spelling mistakes.
10. Tables 2-5 should be simplified, the "Amount found" column can be removed.
11. The authors should also consider moving some figures into the electronic supplementary material file.

Review form: Reviewer 3

Is the manuscript scientifically sound in its present form?

No

Are the interpretations and conclusions justified by the results?

Yes

Is the language acceptable?

Yes

Do you have any ethical concerns with this paper?

No

Have you any concerns about statistical analyses in this paper?

No

Recommendation?

Major revision is needed (please make suggestions in comments)

Comments to the Author(s)

Comments:

1. In the title the authors mentioned that the method was applied to human plasma. However, only spiked plasma appeared in the manuscript and no real plasma was analyzed. Analysis of spiked plasma is not a guarantee for the successful application in real samples. So the authors should try to analyze real plasma, otherwise the plasma should be removed from the title.
2. Page 5, line 23: the authors said "However, its direct application is inadequate for the assay of multi-component samples;" This statement is not true because there are many mixtures can be analyzed using direct measurement. So this statement should be modified.
3. Page 5, line 39 – 53: The authors explain the difference between the two modes of synchronous spectrofluorimetry. Actually there is no difference between the two mode. In the first one use constant wavelength difference while the second use constant energy difference in cm^{-1} . Both the wavelength and wave number are inter-related as the wave number is the

reciprocal of wavelength (or 1/wavelength). So both type are the same. In all synchronous modes only the wavelength difference mode is used. There is mean for using the energy difference mode. It is wasting the time and efforts for repeating the same work and obtaining the same results as constant wavelength difference mode. In addition, the energy difference has no advantage over the other mode. In contrast to the claimed advantage for CESS method (mentioned in page 15, line 33), the sensitivity of both methods are almost the same LODs for FEB is 0.40 and 0.25 ng mL⁻¹ while for IBU the values are 5.6 and 3.4 ng mL⁻¹ for the first method and second methods, respectively.

4. Page 7, line 51-56: The authors said "Afterwards, SF spectra were recorded at 329 nm for FEB and 258 nm for IBU, in the range of 200-400 nm keeping a constant $\Delta \lambda$ of 40 nm between excitation and emission monochromators ". This sentence is confusing and should be modified. The phrase ", in the range of 200-400 nm " can be deleted .
5. Page 8, line 35: under the title "Analysis of FEB and IBU in tablets " and page 14, line 20 – 33, the authors described the analysis of separate dosage forms but not simultaneous determination . This is not matched with the title of the manuscript.
6. According to Figure 2, it very easy to determine FEB in the presence of IBU as there is not interference of IBU spectra with that of FEB. So there is no need for the synchronous mode for FEB analysis.
7. Page 11, line 50-53: The authors said "The studied surfactants include β -cyclodextrin, cetrimide, sodium dodecyl sulphate, tween-80, and carboxy methyl cellulose. ". Actually β -cyclodextrin and carboxy methyl cellulose are not surfactants.
8. Page 12, line 21 – 39: There is no need to present the equations as all the constants in these equations are mentioned in Table 1.
9. Page 13, lines 39 – 54 : The title section " 3.5.5. Selectivity " do not contain a new data, all results are previously mentioned and there is no additional experiment was carried to test selectivity of the method.
10. The number of figure should be reduced and the layout of the figure should be improved.
11. Table are presented in a very bad formats, the data should be reduced by deleting the individual numbers. In tables 2,3 and 4, the mean and SD should be calculated for each concentration but not for the data obtained from different concentrations.
12. About the greenness of the methods: All spectrofluorimetric methods are green specially if the used solvent is water and therefore, there is no need to apply GAPI or analytical Eco-Scale.

Decision letter (RSOS-210354.R0)

Dear Dr Magdy:

Title: Two Different Synchronous Spectrofluorimetric Approaches for Simultaneous Determination of Febuxostat and Ibuprofen: Application to Human Plasma
 Manuscript ID: RSOS-210354

The editor assigned to your manuscript has now received comments from reviewers. We would like you to revise your paper in accordance with the referee and Subject Editor suggestions which can be found below (not including confidential reports to the Editor). Please note this decision does not guarantee eventual acceptance.

Please submit your revised paper before 02-May-2021. Please note that the revision deadline will expire at 00.00am on this date. If we do not hear from you within this time then it will be assumed that the paper has been withdrawn. In exceptional circumstances, extensions may be possible if agreed with the Editorial Office in advance. We do not allow multiple rounds of revision so we urge you to make every effort to fully address all of the comments at this stage. If deemed necessary by the Editors, your manuscript will be sent back to one or more of the original reviewers for assessment. If the original reviewers are not available we may invite new reviewers.

RSC Associate Editor:
Comments to the Author:
(There are no comments.)

RSC Subject Editor:
Comments to the Author:
(There are no comments.)

Reviewers' Comments to Author:

Reviewer: 1

Comments to the Author(s)

In this study, the authors developed two green and sensitive synchronous spectrofluorometric methods (CWSS and CESS) for simultaneous determination of febuxostat and ibuprofen. They applied the proposed methods to pharmaceuticals and human plasma. It is shown how green the methods are using two different greenness assessment tools; GAPI and analytical Eco-Scale. I think the developed methods are useful and have advantages over the published ones.

The manuscript is well written and the experimental strategies were explained in detail. The context of the subject in the literature is demonstrated and the paper is easy to understand. I see this manuscript is acceptable for publication after few minor changes. The suggestions were listed below.

1. The manuscript should be rechecked and any typos have to be corrected.
2. In abstract: line number 33, "Furthermore" may be removed.
3. In section 2.2, the used cell should be written.
4. In section 3.2: line number 18, change "The intensities of SF" into "The intensities of SF spectra".
5. Reference number 60, authors' names need corrections and the title should be consistent with that in Dryad Digital Repository.
6. Table 1: add superscript letters on LOD and LOQ in the table and its footnotes.
7. Table 4: change "pharmaceutical preparations" into "tablet dosage forms" in the caption.

Reviewer: 2

Comments to the Author(s)

The manuscript presents two synchronous spectrofluorimetric approaches for the simultaneous analysis of combined drugs; febuxostat and ibuprofen. The developed methods were applied to pharmaceutical tablets and human plasma. I recommend that the proposed methods are green and sensitive enough for simultaneous estimation of such a mixture. The manuscript is well-written and the topic is interesting. It can be accepted for publication in Royal Society Open Science after performing the required revision.

Comments:

1. Please revise the adjustment of the text within the manuscript to be uniform.
2. In the abstract; "the method greenness" in lines no. 17 and 36 should be corrected into "the methods' greenness".
3. In the Introduction section; the first sentence "Gout is a type of inflammatory arthritis that is triggered by the deposition of monosodium urate crystals in the bones, joints, and parenchymal organs such as the kidney and is frequently associated with hyperuricemia" should be shorter.
4. Section 2.1; the second sentence, change "FEB" into "Febuxostat" so that it doesn't start with an abbreviation.
5. Section 2.2; please add other instruments used for centrifugation, vortex, and filtration.
6. Section 2.4.3; please explain why 10 min sonication is necessary.
7. Section 3.7; line no. 16, "direct methods", remove "methods" as it is a redundant word.
8. Please revise the significant figures all over the manuscript and they should be unified.
9. Please revise the manuscript for grammatical and spelling mistakes.
10. Tables 2-5 should be simplified, the "Amount found" column can be removed.
11. The authors should also consider moving some figures into the electronic supplementary material file.

Reviewer: 3

Comments to the Author(s)

Comments:

1. In the title the authors mentioned that the method was applied to human plasma. However, only spiked plasma appeared in the manuscript and no real plasma was analyzed. Analysis of

- spiked plasma is not a guarantee for the successful application in real samples. So the authors should try to analyze real plasma, otherwise the plasma should be removed from the title.
2. Page 5, line 23: the authors said "However, its direct application is inadequate for the assay of multi-component samples;" This statement is not true because there are many mixtures can be analyzed using direct measurement. So this statement should be modified.
 3. Page 5, line 39 – 53: The authors explain the difference between the two modes of synchronous spectrofluorimetry. Actually there is no difference between the two mode. In the first one use constant wavelength difference while the second use constant energy difference in cm^{-1} . Both the wavelength and wave number are inter-related as the wave number is the reciprocal of wavelength (or $1/\text{wavelength}$). So both type are the same. In all synchronous modes only the wavelength difference mode is used. There is mean for using the energy difference mode. It is wasting the time and efforts for repeating the same work and obtaining the same results as constant wavelength difference mode. In addition, the energy difference has no advantage over the other mode. In contrast to the claimed advantage for CESS method (mentioned in page 15, line 33), the sensitivity of both methods are almost the same LODs for FEB is 0.40 and 0.25 ng mL^{-1} while for IBU the values are 5.6 and 3.4 ng mL^{-1} for the first method and second methods, respectively.
 4. Page 7, line 51-56: The authors said "Afterwards, SF spectra were recorded at 329 nm for FEB and 258 nm for IBU, in the range of 200-400 nm keeping a constant $\Delta \lambda$ of 40 nm between excitation and emission monochromators ". This sentence is confusing and should be modified. The phrase ", in the range of 200-400 nm " can be deleted .
 5. Page 8, line 35: under the title "Analysis of FEB and IBU in tablets " and page 14, line 20 – 33, the authors described the analysis of separate dosage forms but not simultaneous determination . This is not matched with the title of the manuscript.
 6. According to Figure 2, it very easy to determine FEB in the presence of IBU as there is not interference of IBU spectra with that of FEB. So there is no need for the synchronous mode for FEB analysis.
 7. Page 11, line 50-53: The authors said "The studied surfactants include β -cyclodextrin, cetrimide, sodium dodecyl sulphate, tween-80, and carboxy methyl cellulose. ". Actually β -cyclodextrin and carboxy methyl cellulose are not surfactants.
 8. Page 12, line 21 – 39: There is no need to present the equations as all the constants in these equations are mentioned in Table 1.
 9. Page 13, lines 39 – 54 : The title section " 3.5.5. Selectivity " do not contain a new data, all results are previously mentioned and there is no additional experiment was carried to test selectivity of the method.
 10. The number of figure should be reduced and the layout of the figure should be improved.
 11. Table are presented in a very bad formats, the data should be reduced by deleting the individual numbers. In tables 2,3 and 4, the mean and SD should be calculated for each concentration but not for the data obtained from different concentrations.
 12. About the greenness of the methods: All spectrofluorimetric methods are green specially if the used solvent is water and therefore, there is no need to apply GAPI or analytical Eco-Scale.

Author's Response to Decision Letter for (RSOS-210354.R0)

See Appendix A.

RSOS-210354.R1 (Revision)

Review form: Reviewer 1

Is the manuscript scientifically sound in its present form?

Yes

Are the interpretations and conclusions justified by the results?

Yes

Is the language acceptable?

Yes

Do you have any ethical concerns with this paper?

No

Have you any concerns about statistical analyses in this paper?

No

Recommendation?

Accept as is

Comments to the Author(s)

All corrections have been adequately made and can be published in its current format

Decision letter (RSOS-210354.R1)

Dear Dr Magdy:

Title: Two Different Synchronous Spectrofluorimetric Approaches for Simultaneous Determination of Febuxostat and Ibuprofen

Manuscript ID: RSOS-210354.R1

It is a pleasure to accept your manuscript in its current form for publication in Royal Society Open Science. The chemistry content of Royal Society Open Science is published in collaboration with the Royal Society of Chemistry.

RSC Associate Editor:
Comments to the Author:
(There are no comments.)

RSC Associate Editor:
Comments to the Author:
(There are no comments.)

Reviewer(s)' Comments to Author:
Reviewer: 1

Comments to the Author(s)
All corrections have been adequately made and can be published in its current format

Dear Prof. Editor of Royal Society Open Science,

On the behalf of all authors, I would like to thank you for the opportunity that we have been given to further revise our manuscript entitled: "**Two Different Synchronous Spectrofluorimetric Approaches for Simultaneous Determination of Febuxostat and Ibuprofen: Application to Human Plasma**". We appreciate the time and effort that you and the reviewers have dedicated to providing your valuable feedback and insightful comments on the manuscript.

We have carefully revised the manuscript taking into account your recommendations and the reviewers' comments. Please, find enclosed the revised version of the manuscript where the areas containing the major changes were highlighted and the font color was changed into red.

Here is a point-by-point response to the reviewers' comments and concerns.

Comments from Reviewer #1

In this study, the authors developed two green and sensitive synchronous spectrofluorometric methods (CWSS and CESS) for simultaneous determination of febuxostat and ibuprofen. They applied the proposed methods to pharmaceuticals and human plasma. It is shown how green the methods are using two different greenness assessment tools; GAPI and analytical Eco-Scale. I think the developed methods are useful and have advantages over the published ones. The manuscript is well written and the experimental strategies were explained in detail. The context of the subject in the literature is demonstrated and the paper is easy to understand. I see this manuscript is acceptable for publication after few minor changes. The suggestions were listed below.

1. The manuscript should be rechecked and any typos have to be corrected

Author's reply: The manuscript was carefully revised and any grammatical or spelling mistakes were corrected as recommended by the reviewer.

2. In abstract: line number 33, "Furthermore" may be removed.

Author's reply: The required modification was performed as recommended.

3. In section 2.2, the used cell should be written.

Author's reply: Thank you for pointing this out. The used cell was written in the "2.2" section as recommended.

4. In section 3.2: line number 18, change “The intensities of SF” into “The intensities of SF spectra”.

Author's reply: It has been changed as recommended.

5. Reference number 60, authors' names need corrections and the title should be consistent with that in Dryad Digital Repository.

Author's reply: Reference number 60 was revised and corrected according to the reviewer's comment.

6. Table 1: add superscript letters on LOD and LOQ in the table and its footnotes.

Author's reply: Superscript letters were added on LOD and LOQ in Table 2 and its footnotes as recommended.

7. Table 4: change “pharmaceutical preparations” into “tablet dosage forms” in the caption.

Author's reply: It has been changed as recommended.

Comments from Reviewer #2

The manuscript presents two synchronous spectrofluorimetric approaches for the simultaneous analysis of combined drugs; febuxostat and ibuprofen. The developed methods were applied to pharmaceutical tablets and human plasma. I recommend that the proposed methods are green and sensitive enough for simultaneous estimation of such a mixture. The manuscript is well-written and

the topic is interesting. It can be accepted for publication in Royal Society Open Science after performing the required revision.

1. Please revise the adjustment of the text within the manuscript to be uniform.

Author's reply: The reviewer's comment was considered during the preparation of the revised manuscript. The adjustment of the text was carefully revised and unified throughout the manuscript.

2. In the abstract; “the method greenness” in lines no. 17 and 36 should be corrected into “the methods' greenness”.

Author's reply: It has been corrected as recommended by the reviewer.

3. In the Introduction section; the first sentence “Gout is a type of inflammatory arthritis that is triggered by the deposition of monosodium urate crystals in the bones, joints, and parenchymal organs such as the kidney and is frequently associated with hyperuricemia” should be shorter

Author's reply: The required modification was performed. The mentioned sentence was shortened and divided into two short sentences in response to the reviewer's comment.

4. Section 2.1; the second sentence, change “FEB” into “Febuxostat” so that it doesn't start with an abbreviation.

Author's reply: It has been changed as recommended.

5. Section 2.2; please add other instruments used for centrifugation, vortex, and filtration.

Author's reply: Thank you for your valuable comment. The instruments used for centrifugation, vortex, and filtration were added under the "2.2" section as recommended.

6. Section 2.4.3; please explain why 10 min sonication is necessary.

Author's reply: For analysis of FEB and IBU in commercial tablets, 10 min. sonication was necessary after the addition of 40 mL methanol to ensure complete extraction and separation of the cited drugs from the tablet excipients which are insoluble in methanol. This in turn guarantees the method selectivity for the determination of the drugs in presence of tablet excipients with high recoveries and without any interference; which is in compliance with the pharmacopeial methods for assay of drugs in pharmaceutical preparations. This method was reported in many published articles [1-4].

References:

[1] Abo El Abass, S., Elmansi, H. 2018 Synchronous fluorescence as a green and selective tool for simultaneous determination of bambuterol and its main degradation product, terbutaline. *R. Soc. Open Sci.* **5**, 181359.

[2] Elmansi, H., Roshdy, A., Shalan, S., El-Brashy, A. 2020 Combining derivative and synchronous approaches for simultaneous spectrofluorimetric determination of terbinafine and itraconazole. *R. Soc. Open Sci.* **7**, 200571.

[3] Eman, I., Ragab, M. A. 2018 Derivative synchronous spectrofluorimetry: application to the analysis of two binary mixtures containing codeine in dosage forms. *Spectrochim. Acta A.* **204**, 677-684.

[4] Maher, H. M. 2008 Simultaneous determination of naproxen and diflunisal using synchronous luminescence spectrometry. *J. Fluorescence.* **18**, 909-917.

7. Section 3.7; line no. 16, “direct methods”, remove “methods” as it is a redundant word.

Author's reply: The word “methods” was removed as recommended by the reviewer.

8. Please revise the significant figures all over the manuscript and they should be unified.

Author's reply: The significant figures were carefully revised and unified throughout the manuscript in response to the reviewer's comment.

9. Please revise the manuscript for grammatical and spelling mistakes.

Author's reply: The manuscript was carefully revised and any grammatical or spelling mistakes were corrected as recommended.

10. Tables 2-5 should be simplified, the "Amount found" column can be removed.

Author's reply: Tables 2, 3, 4, and 5 were simplified and the "Amount found" column was removed as recommended by the second and third reviewers.

11. The authors should also consider moving some figures into the electronic supplementary material file.

Author's reply: Thank you for your valuable suggestion. Figure 7 was moved to the electronic supplementary material file as recommended by the second and third reviewers, then the figures were renumbered in the revised manuscript.

Comments from Reviewer #3

1. In the title the authors mentioned that the method was applied to human plasma. However, only spiked plasma appeared in the manuscript and no real plasma was analyzed. Analysis of spiked plasma is not a guarantee for the successful application in real samples. So the authors should try to analyze real plasma, otherwise the plasma should be removed from the title.

Author's reply: Thank you for your valuable suggestion. Plasma was removed from the title as recommended and the title of the revised manuscript was changed into "Two Different Synchronous Spectrofluorimetric Approaches for Simultaneous Determination of Febuxostat and Ibuprofen".

2. Page 5, line 23: the authors said "However, its direct application is inadequate for the assay of multi-component samples; " This statement is not true because there are many mixtures can be analyzed using direct measurement. So this statement should be modified.

Author's reply: Thank you for pointing this out. It is true that the direct emission spectrofluorimetric technique could be applied for the determination of some mixtures, but this can occur if there is an adequate resolution between the mixture components and well-resolved peaks which allow the determination of each component in presence of the other without interference. Selectivity problems can arise if the components have broad bands and overlapped emission spectra so that resolution between mixture components is insufficient. The limited selectivity of direct emission spectrofluorimetry can be improved by applying total luminescence, synchronous spectrofluorimetric, or derivative synchronous spectrofluorimetric techniques [1,2]. This is what we meant by the word "inadequate".

The statement was modified in response to the reviewer's comment into "However, selectivity problems can arise during the simultaneous determination of multi-component samples with broad bands and overlapped emission spectra".

References:

- [1] A. Andrade-Eiroa, G. de-Armas, J.-M. Estela, V. Cerda, Critical approach to synchronous spectrofluorimetry. I, *Tr. Anal. Chem.* **29** (2010) 885-901.
- [2] Andrade-Eiroa, A., de-Armas, G., Estela, J.-M., Cerda, V. 2010 Critical approach to synchronous spectrofluorimetry. II. *Tr. Anal. Chem.* **29**, 902-927.

3. Page 5, line 39-53: The authors explain the difference between the two modes of synchronous spectrofluorimetry. Actually there is no difference between the two modes. In the first one use constant wavelength difference while the second use constant energy difference in cm^{-1} . Both the wavelength and wave number are inter-related as the wave

number is the reciprocal of wavelength (or 1/wavelength). So both types are the same. In all synchronous modes only the wavelength difference mode is used. There is mean for using the energy difference mode. It is wasting the time and efforts for repeating the same work and obtaining the same results as constant wavelength difference mode. In addition, the energy difference has no advantage over the other mode. In contrast to the claimed advantage for CESS method (mentioned in page 15, line 33), the sensitivity of both methods are almost the same LODs for FEB is 0.40 and 0.25 ng mL⁻¹ while for IBU the values are 5.6 and 3.4 ng mL⁻¹ for the first method and second methods, respectively.

Author's reply: Thank you for your valuable comment. There are different modes of synchronous spectrofluorimetry including constant-wavelength synchronous spectrofluorimetry (CWSS), constant-energy synchronous spectrofluorimetry (CESS), variable-angle synchronous spectrofluorimetry (VASS) [variable-angle wavelength (VAW) and variable-angle energy (VAE)], and matrix isopotential synchronous spectrofluorimetry (MISS). The differences between these modes, their principles, and applications have been detailed in the literature [1-3].

It is true that CWSS is the most used technique as the major advantage of CWSS is that it can be facilely conducted using the existing capabilities of data acquisition software programs of most commercial fluorescence spectrometers [1]. Therefore, most published reports use CWSS but it is not the only mode to be applied. Synchronous scanning could also be carried out in CESS mode since 1982, when Inman and Winefordner [4] developed this mode, which has demonstrated great utility in the resolution of multi-component mixtures [5]. Since this date, CESS was applied and reported in some published articles [6-10].

It was reported that, CESS could offer improvement in sensitivity and selectivity and overcome the limitations of CWSS technique [11,12]. This is because CESS can greatly reduce the effect of solvent Raman scatter and Rayleigh scatter interferences with the fluorescence spectra of the analytes [12]. Conventional CESS does not entirely eliminate Rayleigh and Raman interference especially when the difference between the excitation and emission wavelengths exceeds the

difference between the wavelengths of the Rayleigh and Raman peaks [12,13]. This advantage of CESS could be useful especially in the case of weakly fluorescent analytes or low concentrations of strongly fluorescent compounds [12,13].

Some published articles used both CWSS and CESS in the same study and reported an improvement of sensitivity in the case of CESS over CWSS [9, 10,13]. Similarly, the current study applied both techniques to obtain sensitivity as high as possible. It was found that CESS improved the sensitivity of the developed method to some extent where LODs decreased to almost half which could be important for the determination of very low concentrations of the studied drugs.

Hopefully, the point is clarified.

References:

- [1] Samokhvalov, A. 2020 Analysis of various solid samples by synchronous fluorescence spectroscopy and related methods: A review. *Talanta*. **216**, 120944.
- [2] Andrade-Eiroa, A., de-Armas, G., Estela, J.-M., Cerda, V. 2010 Critical approach to synchronous spectrofluorimetry. I. *Tr. Anal. Chem.* **29**, 885-901.
- [3] Andrade-Eiroa, A., de-Armas, G., Estela, J.-M., Cerda, V. 2010 Critical approach to synchronous spectrofluorimetry. II. *Tr. Anal. Chem.* **29**, 902-927.
- [4] Inman, E. L., Winefordner, J. D. 1982 Constant energy synchronous fluorescence for analysis of polynuclear aromatic hydrocarbon mixtures. *Anal. Chem.* **54**, 2018-2022.
- [5] Inman, E. L., Files, L. A., Winefordner, J. D. 1986 Theoretical optimization of parameter selection in constant energy synchronous luminescence spectrometry. *Anal. Chem.* **58**, 2156-2160.
- [6] Eman, I., Ragab, M. A. 2018 Derivative synchronous spectrofluorimetry: application to the analysis of two binary mixtures containing codeine in dosage forms. *Spectrochim. Acta A*. **204**, 677-684.

- [7] Maher, H. M. 2008 Simultaneous determination of naproxen and diflunisal using synchronous luminescence spectrometry. *J. Fluorescence*. **18**, 909-917.
- [8] Eiroa, A. A., Blanco, E. V., Mahia, P. L., Lorenzo, S. M., Rodriguez, D. P., Fernández, E. F. 2000 Determination of polycyclic aromatic hydrocarbons (PAHs) in a complex mixture by second-derivative constant-energy synchronous spectrofluorimetry. *Talanta*. **51**, 677-684.
- [9] Pulgarín, J. M., Molina, A. A. 1994 Determination of nafcillin and methicillin by different spectrofluorimetric techniques. *Talanta*. **41**, 21-30.
- [10] Nevado, J. B., Pulgarín, J. M., Laguna, M. G. 1995 Simultaneous determination of pyridoxal and pyridoxamine by different spectrofluorimetric techniques. *Talanta*. **42**, 129-136.
- [11] Eiroa, A. A., Huckins, S., Blanco, E. V., Mahia, P. L., Lorenzo, S. M., Rodríguez, D. P. 2000 Optimizing resolution in constant-energy synchronous spectrofluorimetry. *Appl. Spectrosc.* **54**, 1534-1538.
- [12] Inman Jr, E. L., Winefordner, J. D. 1982 Constant-energy synchronous fluorescence for reduction of Raman scatter interference. *Anal. Chim. Acta*. **138**, 245-252.
- [13] Andre, J., Bouchy, A., Jezequel, J. 1986 Increased apparent sensitivity by means of constant-energy synchronous spectrofluorimetry. *Anal. Chim. Acta*. **185**, 91-99.

4. Page 7, line 51-56: The authors said "Afterwards, SF spectra were recorded at 329 nm for FEB and 258 nm for IBU, in the range of 200-400 nm keeping a constant $\Delta\lambda$ of 40 nm between excitation and emission monochromators ". This sentence is confusing and should be modified. The phrase ", in the range of 200-400 nm" can be deleted.

Author's reply: The phrase "in the range of 200-400 nm" was deleted and the sentence was modified into "Afterwards, SF spectra were recorded at 329 and 258 nm for FEB and IBU, respectively keeping a constant $\Delta\lambda$ of 40 nm between excitation and emission monochromators" as recommended by the reviewer.

5. Page 8, line 35: under the title "Analysis of FEB and IBU in tablets " and page 14, line 20 - 33, the authors described the analysis of separate dosage forms but not simultaneous determination. This is not matched with the title of the manuscript.

Author's reply: Thank you for your comment. The proposed methods were applied for the simultaneous determination of FEB and IBU in synthetic mixtures of pure forms and spiked human plasma samples which is consistent with the title of the manuscript. We want to clarify that, FEB and IBU are co-administered drugs for the treatment of gout as mentioned in the "Introduction" section and not co-formulated so that we applied the developed methods for the determination of each drug in its tablet dosage form as reported in many published articles [1-3].

[1] Elmansi, H., Roshdy, A., Shalan, S., El-Brashy, A. 2020 Combining derivative and synchronous approaches for simultaneous spectrofluorimetric determination of terbinafine and itraconazole. *R. Soc. Open Sci.* **7**, 200571.

[2] El Sharkasy, M. E., Walash, M., Belal, F., Salim, M. 2020 First derivative synchronous spectrofluorimetric method for the simultaneous determination of propofol and cisatracurium besylate in biological fluids. *Luminescence.* **35**, 312-320.

[3] El Sharkasy, M. E., Walash, M., Belal, F., Salim, M. 2020 Conventional and first derivative synchronous spectrofluorimetric methods for the simultaneous determination of cisatracurium and nalbuphine in biological fluids. *Spectrochim. Acta A.* **228**, 117841.

6. According to Figure 2, it very easy to determine FEB in the presence of IBU as there is not interference of IBU spectra with that of FEB. So there is no need for the synchronous mode for FEB analysis.

Author's reply: Thank you for pointing this out. Although the emission spectra of FEB and IBU are not overlapped as shown in Figure 2, we preferred to use synchronous spectrofluorimetry to be able to determine both drugs simultaneously in a single scan. The use of conventional emission

spectrofluorimetry for analysis of such mixture will be tedious and time-consuming where each mixture will need to be measured two times; one at $\lambda_{\text{excitation}}$ of FEB and the other at $\lambda_{\text{excitation}}$ of IBU. This reason in addition to the well-known advantages of the synchronous spectrofluorimetric technique which was discussed in detail under the "3.1" section made the synchronous spectrofluorimetric technique the best choice for the analysis of such a binary mixture.

7. Page 11, line 50-53: The authors said "The studied surfactants include β -cyclodextrin, cetrimide, sodium dodecyl sulphate, tween-80, and carboxy methyl cellulose. ". Actually β -cyclodextrin and carboxy methyl cellulose are not surfactants.

Author's reply: Thank you for your valuable comment. It is true that β -cyclodextrin and carboxy methyl cellulose are not surfactants. They are macromolecules and their mechanism as organized media is similar to surfactants. They provide a viscous and very rigid microenvironment that can inhibit quenching by molecular oxygen and restrict the freedom of fluorophores and consequently diminish the probabilities of non-radiative processes. These factors can increase the fluorescence quantum yield and enhance the fluorescence intensity [1,2]. Therefore, the word "surfactants" was replaced by "organized media" in the revised manuscript according to the reviewer's comment. In addition, the sentence was corrected into "The studied organized media include surfactants such as cetrimide, sodium dodecyl sulphate, and tween-80, as well as macromolecules such as carboxy methyl cellulose, and β -cyclodextrin" in "3.4.3." section. We used the mentioned examples for organized media in the study as reported in most of the published articles [1-6].

References:

[1] El Sharkasy, M. E., Walash, M., Belal, F., Salim, M. 2020 Conventional and first derivative synchronous spectrofluorimetric methods for the simultaneous determination of cisatracurium and nalbuphine in biological fluids. *Spectrochim. Acta A*. **228**, 117841.

[2] Atia, N. N., El-Gizawy, S. M., Hosny, N. M. 2019 Facile micelle-enhanced spectrofluorimetric method for picogram level determination of febuxostat; application in tablets and in real human plasma. *Microchem. J.* **147**, 296-302.

[3] El Sharkasy, M. E., Walash, M., Belal, F., Salim, M. 2020 First derivative synchronous spectrofluorimetric method for the simultaneous determination of propofol and cisatracurium besylate in biological fluids. *Luminescence.* **35**, 312-320.

[4] El Gamal, R., El Abass, S. A., Elmansi, H. M. 2020 Quick simultaneous analysis of bambuterol and montelukast based on synchronous spectrofluorimetric technique. *R. Soc. Open Sci.* **7**, 201156.

[5] Abo El Abass, S., Elmansi, H. 2018 Synchronous fluorescence as a green and selective tool for simultaneous determination of bambuterol and its main degradation product, terbutaline. *R. Soc. Open Sci.* **5**, 181359.

[6] Elmansi, H., Roshdy, A., Shalan, S., El-Brashy, A. 2020 Combining derivative and synchronous approaches for simultaneous spectrofluorimetric determination of terbinafine and itraconazole. *R. Soc. Open Sci.* **7**, 200571.

8. Page 12, line 21 – 39: There is no need to present the equations as all the constants in these equations are mentioned in Table 1.

Author's reply: The equations were removed as recommended by the reviewer.

9. Page 13, lines 39 – 54 : The title section " 3.5.5. Selectivity " do not contain a new data, all results are previously mentioned and there is no additional experiment was carried to test selectivity of the method.

Author's reply: Thank you for your valuable comment. The ability of the developed methods to determine each drug in presence of the other without interference as well as their concurrent analysis in complicated biological matrices of spiked human plasma samples with high % recoveries (97.3-101.9 %) and low % RSD values (less than 2 %) proved the high selectivity of the

developed methods. The proposed methods were also efficiently applied for the estimation of FEB and IBU in their tablets without interference from co-formulated excipients. Therefore, no additional experiment was carried out to test the selectivity of the methods.

10. The number of figures should be reduced and the layout of the figures should be improved.

Author's reply: The number of figures was reduced as recommended by the second and third reviewers. Figure 7 was moved to the electronic supplementary material file, and then the figures were renumbered in the revised manuscript. The layout and resolution of the figures were also improved as recommended.

11. Tables are presented in very bad formats, the data should be reduced by deleting the individual numbers. In tables 2,3 and 4, the mean and SD should be calculated for each concentration but not for the data obtained from different concentrations.

Author's reply: All tables were carefully revised and their formats were improved. The tables were modified as follows:

- The data were reduced, where the “Amount found” column and the individual numbers were removed from Tables 2, 3, 4, and 5 as recommended by the second and third reviewers.
- The mean and SD were calculated for each concentration and added to the mentioned tables as recommended.

12. About the greenness of the methods: All spectrofluorimetric methods are green specially if the used solvent is water and therefore, there is no need to apply GAPI or analytical Eco-Scale.

Author's reply: Thank you for your valuable comment. It is true that most of the spectrofluorimetric methods are green but not all methods. The use of water as a diluting solvent increases the method greenness but there are other factors that can affect the greenness of a

spectrofluorimetric method. In GAPI, many parameters are considered including; sample preparation (collection, preservation, transport, storage, type of method; direct or indirect, scale of extraction, solvents/reagents used, and additional treatments), reagent and solvents (amount, health hazard, and safety hazard), instrumentation (energy, occupational hazard, waste amount, and waste treatment), and whether the method is for qualification and quantification or for qualification only [1]. In analytical Eco-Scale, a numerical score is calculated indicating the greenness of the method, where penalty points are subtracted from a total of 100 points for each negative impact on the environment (e.g. hazardous chemicals used, waste generation, and high energy consumption) [2]. Therefore, if any spectrofluorimetric method doesn't fulfill most of these requirements, its greenness score may be insufficient. For example, in some spectrofluorimetric methods, derivatization of the studied drug using a hazardous reagent and drastic conditions or extraction using an organic solvent may be required which can significantly affect its greenness. Hence, each method should be evaluated for greenness taking into account all experimental parameters. Similarly, we preferred to evaluate the green profile of the developed methods using two reported greenness assessment tools; GAPI and analytical Eco-Scale to consider all the experimental parameters and confirm that they are sufficiently green.

References:

- [1] Płotka-Wasyłka, J. 2018 A new tool for the evaluation of the analytical procedure: Green Analytical Procedure Index. *Talanta*. **181**, 204-209.
- [2] Gałuszka, A., Migaszewski, Z. M., Konieczka, P., Namieśnik, J. 2012 Analytical Eco-Scale for assessing the greenness of analytical procedures. *Tr. Anal. Chem.* **37**, 61-72.

We hope that we covered the most suggestions and recommendations from the reviewers and our revised manuscript will be acceptable for publication.

Sincerely,

Galal Magdy